# Prospects and Challenges in the Development of Universal Influenza Vaccines

**DOI:** 10.3390/vaccines8030361

**Published:** 2020-07-06

**Authors:** Anders Madsen, Rebecca Jane Cox

**Affiliations:** 1Influenza Centre, Department of Clinical Science, University of Bergen, 5021 Bergen, Norway; Anders.Madsen@student.uib.no; 2Department of Microbiology, Haukeland University Hospital, 5021 Bergen, Norway

**Keywords:** influenza, vaccine, next-generation vaccine

## Abstract

Current influenza vaccines offer suboptimal protection and depend on annual reformulation and yearly administration. Vaccine technology has rapidly advanced during the last decade, facilitating development of next-generation influenza vaccines that can target a broader range of influenza viruses. The development and licensure of a universal influenza vaccine could provide a game changing option for the control of influenza by protecting against all influenza A and B viruses. Here we review important findings and considerations regarding the development of universal influenza vaccines and what we can learn from this moving forward with a SARS-CoV-2 vaccine design.

## 1. Introduction

An influenza virus infection typically manifests as a fever, sore throat, cough, runny nose, myalgia, and headaches. People in certain risk groups (the elderly, children <59 months of age, pregnant women, and individuals with chronic conditions) have a higher chance of developing more severe illness with complications such as fulminant pneumonia. Seasonal influenza causes around 290,000 to 650,000 respiratory deaths each year [1]. However, this number can increase dramatically when, at unpredictable intervals, influenza pandemics occur. These global epidemics are characterized by the emergence of a novel influenza virus to which most humans are immunologically naïve [2]. The emergence of the A/H1N1pdm09 virus (“swine flu”) was the first influenza pandemic of the 21st century. This H1N1 virus has subsequently continued to circulate in the human population as a seasonal virus, together with influenza A/H3N2 and influenza B (Yamagata-like and Victoria-like) viruses. The current seasonal influenza vaccines contain strains from each of these viruses. The World Health Organization (WHO) decides biannually which strains should be included in the vaccine based on which viruses are most likely to circulate in the upcoming season. Annual seasonal influenza vaccination remains the most cost-effective measure to reduce burden of the disease. However, the currently licensed influenza vaccines lack two essential attributes: Firstly, they do not produce durable protective immunity. Secondly, they do not produce a cross-reactive immune response that can neutralize diverse influenza virus strains. Cross-reactivity is necessary because of the antigenic plasticity of the viral membrane protein hemagglutinin (HA), which leads to antigenically drifted viruses and potential mismatches with the vaccine strains [3]. This shortcoming can be addressed by developing new vaccines targeting the more antigenically conserved regions of the influenza virus [4]. In order to overcome these barriers, efforts have been made to develop next-generation influenza vaccines that provide robust, long-lasting protection against diverse influenza A subtypes (Table 1) [5]. The ultimate goal is to develop a “universal” influenza vaccine that covers all influenza A and B viruses. Here, we will examine the latest research on next-generation vaccine development and discuss the prospects and challenges of developing a universal influenza vaccine.

## 2. Influenza Virus and Target Epitopes

The influenza virus is an enveloped virus with a segmented, negative-sense, single-strand RNA genome (Figure 1). There are four types of influenza: A, B, C, and D. Influenza A and B are the only types that cause seasonal epidemics in humans. Influenza A viruses are further subdivided into subtypes and strains depending on the characteristics of the most abundant surface proteins, HA and neuraminidase (NA).

HA is a trimeric glycoprotein responsible for the attachment of the virus to the surface of the host cell by binding to sialic acid receptors. There are 18 subtypes of HA within the influenza A viruses. These can be divided into two distinct phylogenetic groups: group 1 HA (H1, H2, H5, H6, H8, H9, H11, H12, H13, H16, H17, and H18) and group 2 HA (H3, H4, H7, H10, H14, and H15). The HA protein comprises a head domain and a stalk domain, the latter being more antigenically conserved. Antibodies targeting the HA head domain are usually strain-specific and neutralize influenza viruses by inhibiting the binding of HA to sialic acid receptors on the surface of the host cell. HA stalk-specific antibodies can provide heterosubtypic protection by blocking viral fusion with the host cell and by eliminating infected host cells through antibody-dependent cellular cytotoxicity (ADCC) [7,8]. The stalk-specific antibodies are broadly cross-reactive within an HA group [9,10].

While HA remains the main target of current inactivated influenza vaccines (IIVs), NA has recently been recognized as a potential target for universal vaccines [11,12]. NA functions by catalyzing the cleavage of sialic acids on the surface of an infected cell, leading to the release of newly formed viruses. Several studies have shown that NA antibody titers correlate with protection in humans [13,14,15]. NA-specific antibodies function by inhibiting the enzymatic activity of NA and thus preventing the spread of virions from infected cells. Fc receptor-mediated effector functions such as ADCC have also been reported for NA antibodies [16,17].

The matrix protein 2 ectodomain (M2e) protein functions by transferring H^+^ ions through the viral membrane as the virus is taken up by the endosomes. This leads to conformational changes of the HA molecule and fusion between the viral membrane and the endosome membrane. As a result, the viral RNA is released into the cytosol. M2e-specifc antibodies have been shown to be protective in animal studies, although the exact defense mechanisms are not fully understood. Some studies have reported Fc receptor (FcR)-mediated effector functions by M2e antibodies [18,19,20]. Furthermore, several studies have highlighted the CD4+ and CD8+ T-cell responses induced by M2e [21,22,23,24].

T-cell activation also plays an important role in targeting other conserved proteins that are generally not exposed on the outside of virus particles, such as the nucleoprotein (NP) and matrix protein 1 (M1) [25]. These proteins are highly conserved between influenza A viruses. Influenza-specific interferon-secreting T cells, CD4+ T cells, and CD8+ T cells play an important role in recovery from influenza in humans [26,27,28,29].

## 3. Current Influenza Vaccines

There are currently three types of licensed influenza vaccines: inactivated, live attenuated, and recombinant HA (Figure 2). Inactivated influenza vaccines (IIVs) are the most commonly used, partly because of their well-documented safety and low production costs. The IIVs come in three variants: whole-virion vaccines, split-virion vaccines, and subunit vaccines, which contain purified HA and NA. IIVs are traditionally produced in embryonated chicken eggs. This egg-based approach has several drawbacks: Firstly, the production time is relatively long. Consequently, the producers have to start production months in advance of the flu season, which can lead to mismatches between the vaccine strain and the circulating strain [30]. In addition, the prolonged production time becomes a crucial problem during influenza pandemics, where rapid production and distribution of vaccines are essential. Another issue with the egg-based approach is the occurrence of egg-adapted mutations during the production process, which can lower vaccine effectiveness [31,32]. Cell-based IIVs have been licensed in Europe and the United States as an alternative to the egg-based approach. Another approach for overcoming egg-based vaccine production is the use of recombinant HA vaccines, which are based on a protein-expression system using baculoviruses and insect cells [33]. These vaccines have been licensed in the United States since 2013. However, as with the IIVs, the protection appears to be strain-specific, with limited immunogenicity.

Some vaccine formulations include adjuvants in order to improve immunogenicity and vaccine effectiveness [34]. Adjuvants function by increasing antigen uptake and presentation in the local draining lymph nodes. During the 2009 H1N1 pandemic, the use of the oil-in-water emulsion (AS03) led to higher B and CD4 T-cell responses than vaccination with nonadjuvanted pandemic vaccines [35,36].

In contrast to IIVs and recombinant HA vaccines, the live-attenuated influenza vaccines (LAIV) contain live, cold-adapted viruses that are administered as a nasal spray, leading to a restricted viral replication in the upper respiratory tract of the recipient [37,38]. The LAIV induces broader immune response, including T-cell and mucosal immunity, by mimicking natural infection [39,40,41]. LAIV is currently licensed in North America, Europe, and India. It is generally not recommended for people younger than 2 years of age due to increased risk of wheezing [42] or for immunosuppressed individuals who are at greater risk of severe influenza illness as the vaccine may produce higher virus titers in these individuals, leading to severe side effects.

## 4. Next-Generation Vaccine Platforms

Recent scientific advances have made way for novel vaccine approaches, enabling a more targeted delivery of conserved antigens that can stimulate the innate and adaptive immune systems. Here, we highlight three vaccine platforms that are being used in the development of next-generation influenza vaccines (Figure 2).

Virus-like particles (VLPs) have similar morphological and structural features to viruses but lack the viral genome [43]. They are a useful vaccine component as the immune system recognizes VLPs similarly to viruses but without the risk of replication and recombination. A variety of immunogens have been tested on VLPs, including HA, matrix protein 2 (M2), and NA [44,45,46]. One of the main barriers to the use of this vaccine construct is the challenge of generating sufficient immunogenicity. However, the inclusion of adjuvants such as toll-like receptor ligands has led to improved vaccine effectiveness [47].

In contrast to VLPs, peptide-based vaccines focus on minimal epitopes of the influenza virus, such as T-cell-inducing NP and M1 peptides [48,49]. However, as with VLP-based vaccines, they often require the use of adjuvants or particulate carriers for delivery in order to be sufficiently immunogenic.

Nucleic-acid-based vaccines use RNA- or DNA-sequences for the vaccines antigens. This technology is especially useful in disease outbreaks where rapid vaccine development is needed. Using self-amplifying mRNA, researchers were able to develop a vaccine candidate for the Influenza A H7N9 outbreak in China just 8 days after A/Shanghai/2/2013 (H7N9) sequences were released [50]. This vaccine platform could be a viable option in the ongoing efforts to develop a SARS-CoV-2 vaccine. A phase 1 clinical trial of a DNA vaccine targeting the spike protein of SARS-CoV-1 found that the vaccine was safe and capable of eliciting neutralizing antibodies and cellular immune responses [51].

Viral vector vaccines use carrier viruses such as modified vaccinia virus Ankara (MVA) or adenovirus containing genes expressing influenza-proteins of interest [52]. These vaccines allow for any influenza antigens to be expressed in their native conformation or with modifications, inducing both humoral and cellular immune responses [53,54,55]. Most viral vectors are replication-deficient in mammalian host cells and are therefore safe to use. Issues with pre-existing immunity to some viral vectors have been reported [56], although other vectors, such as MVA, remain immunogenic despite the presence of pre-existing immunity.

## 5. Universal Vaccine Strategies

Influenza vaccines can be divided into two categories based on their immunological features: (1) vaccines providing sterile immunity by eliciting HA head antibodies that prevent viral attachment to sialic acid receptors and (2) vaccines that provide infection-permissive immunity by inducing cellular immunity or broadly reactive antibodies that function in the later stages of the viral life cycle (Figure 3). Many of the infection-permissive vaccines that are in development will probably not be universal vaccine candidates as stand-alone vaccines but could be combined with other approaches to increase immunogenicity.

HA stalk antibodies have emerged as one of the leading strategies in the development of universal vaccine candidates [57]. Several approaches for avoiding the immunodominant HA head domain are under development. One strategy is to remove the globular head domain while maintaining the correct immunogenic conformation of the HA stalk domain [58]. Animal models have shown that vaccination with these “headless” HA proteins can induce heterosubtypic protection within group 1 HA proteins [59,60,61,62]. Other strategies for refocusing the antibody response toward the HA stalk involve maintaining the full-length HA. Krammer and colleagues have developed a method for directing the antibody response to the conserved HA stalk domain by sequential vaccination with chimeric HA proteins consisting of exotic HA head domains to which the recipient is immunologically naïve. The results of a phase 1 clinical trial (NCT03300050) provide evidence that group 1 chimeric HA-based vaccines induce high titers of stalk-reactive immunoglobulin G (IgG) that cross-react to other HA proteins within the same group [63]. A trivalent vaccine comprising group 1, group 2, and influenza B virus chimeric constructs could potentially lead to a universal influenza vaccine.

Advances in the development of monoclonal antibodies (mAbs) have shown that there are cross-reactive epitopes on the HA head domain that can be targeted to give heterosubtypic protection [64,65,66]. A new vaccine approach called mosaic HA has been developed to combine the immunity of conserved epitopes of the HA stalk and head domains. Mosaic HA vaccines are developed by replacing the immunodominant and strain-specific antigenic sites with sequences from exotic HA subtypes. Through sequential immunization, the immune response is refocused toward more cross-reactive immunosubdominant epitopes of the HA stalk and head domains. Cross-reactive HA stalk- and head-specific antibodies can also be induced by computationally optimized, broadly reactive antigens (COBRAs). This approach is based on developing novel HA proteins after multiple rounds of consensus sequence generation from a variety of different HA isolates. Preclinical trials have demonstrated that COBRA-based vaccines provide protection against multiple influenza viruses within a subtype in murine and ferret models [67,68,69].

There has been increased recognition of NA as a target for improved vaccines following the development of practical assays for measuring antibody responses to NA [11]. Although antibodies targeting NA can bind to conserved epitopes and provide heterologous protection [16], they are not really induced by seasonal IIVs. This is mostly due to the amount and stability of the NA in the vaccines. Enhanced NA–antibody responses can be achieved by delivering NA as a supplement to current IIVs or vaccinating with recombinant or viral vector NA in addition to the seasonal IIVs [70,71,72,73,74].

M2e has been an attractive target in the development of the future universal influenza vaccine, as it is highly conserved between influenza viruses [75]. Because of the relatively small size of the M2e, vaccines usually depend on carrier constructs such as VLPs. Preclinical trials have demonstrated that M2e antibodies are protective in animal models, and some M2e vaccines have advanced into clinical trials [76,77]. While the effectiveness of a stand-alone M2e-based vaccine remains limited, recent studies have found promising results by combining an M2e-based vaccine (M2e5x VLP) with other vaccine constructs such as LAIVs [78] and HA VLPs [79].

Most vaccine approaches targeting internal proteins such as M1 and NP involve T-cell activation. T cells are important during an influenza infection because they limit viral replication and shedding by clearing influenza-infected cells. An ongoing phase 3 clinical trial (ClinicalTrials.gov: NCT03450915) is testing a recombinant-peptide based vaccine consisting of conserved epitopes from HA, NP, and M1 [80]. Other vaccine constructs use viral vectors expressing M1 and NP [54,81]. These have been shown to induce CD4+ and CD8+ T cells in clinical trials. Selecting the best approach for inducing T-cell-mediated immunity can be challenging due to the diverse CD4 T-cell repertoire in humans with different degrees of pre-existing immunity. However, with increasing insights into the quantity, functionality, and specificity of CD4 T-cell subsets, there is hope that new correlates of protection can be established in the near future, which would help in selecting the best vaccine candidates for clinical trials [29,82,83]. Some studies evaluating the T-cell response in various infection models have reported immunopathology with excessively large CD8 T-cell responses [84,85]. Other studies have reported a narrowing of T-cell receptors during heterosubtypic challenges [86]. Additional studies that reflect pathogen encounters in humans are needed in order to select the best candidates for next-generation T-cell-based vaccines [87].

## 6. Challenges in the Development of Universal Vaccines

There has been significant progress in the development of novel approaches for universal influenza vaccines. However, additional efforts are needed in order to translate more of these novel approaches into products ready for late-stage clinical trials. Selecting the best candidates for clinical trials is a challenging process for several reasons:(1)There is no uniform evaluation of novel influenza vaccines in preclinical trials. This is partly due to the variety of vaccine platforms and targets. In addition, there is no consensus on which animal model is best suited to accurately reflect human infection [3,88]. This is partly due to the existence of pre-existing immunity in humans, which is difficult to translate into animal models. Other host factors to be considered include age, gender, and chronic diseases, which could affect vaccine efficacy.(2)There are no established correlates of protection for broadly cross-reactive immune responses to influenza. Assays measuring non-HA-head immune responses, such as HA stalk antibodies, NA antibodies, and cellular immunity, are inadequately standardized. Optimization and standardization of such assays are needed in order to assess the protective immunity contributed by broadly reactive or universal vaccines.(3)The funding for universal vaccine development is limited. This leads to down selection of promising candidates before they advance to clinical trials. Regulatory initiatives are needed to enhance advocacy for moving from an annually reformulated vaccine to a universal vaccine, despite the financial risks and inadequate incentives [6].

## 7. Lessons Learned Moving Forward with a SARS-CoV-2 Vaccine Design

As the search for an appropriate SARS-CoV-2 vaccine continues, comparisons can be drawn to influenza vaccine development. Some of the next-generation influenza vaccines use vaccine platforms similar to those used by the SARS-CoV-2 vaccine candidates [89]. Given the urgency of moving forward with a SARS-CoV-2 vaccine, a nucleic-acid-based vaccine would be a convenient approach in order to facilitate rapid production. Although this method has been tested for several influenza vaccine candidates, no DNA or RNA vaccines have yet been licensed for humans. The disadvantages of such vaccines include limited immunogenicity and reports of increased reactogenicity. Another option is the use of viral-vector-based vaccines, which have shown promising results against MERS-CoV and other emerging viruses [90,91,92]. Some viral-vector-based vaccines focusing on the S protein of the SARS-CoV-2 virus are in the preclinical phase. However, it might be years until a nucleic-acid- or viral-vector-based vaccines are licensed for human use since both of these vaccine platforms are novel and have not been extensively tested. Other SARS-CoV-2 vaccines that are currently being tested include recombinant protein vaccines, live-attenuated vaccines, and inactivated vaccines. All of these platforms can be found among the licensed influenza vaccines, each with their own advantages and disadvantages.

In the end, a SARS-CoV-2 vaccine or a universal influenza vaccine can only be successful if implemented effectively [93]. Efforts are needed to improve vaccine uptake by restoring widespread public confidence in vaccines. This is especially important in a pandemic setting. In addition, as the 2009 influenza A/H1N1 pandemic demonstrated, there will likely be an issue of vaccine availability in low- and middle-income countries. Therefore, strategies are needed to ensure sufficient and equitable vaccine distribution to these countries.

## 8. Conclusions

Encouraging progress has been made in the development of universal influenza vaccines. Future universal vaccines will likely consist of a combination of different approaches to induce broad immunity to influenza. Ultimately, the path forward depends on combined action from governments, industries, and the scientific community.

## Figures and Tables

**Figure 1 vaccines-08-00361-f001:**
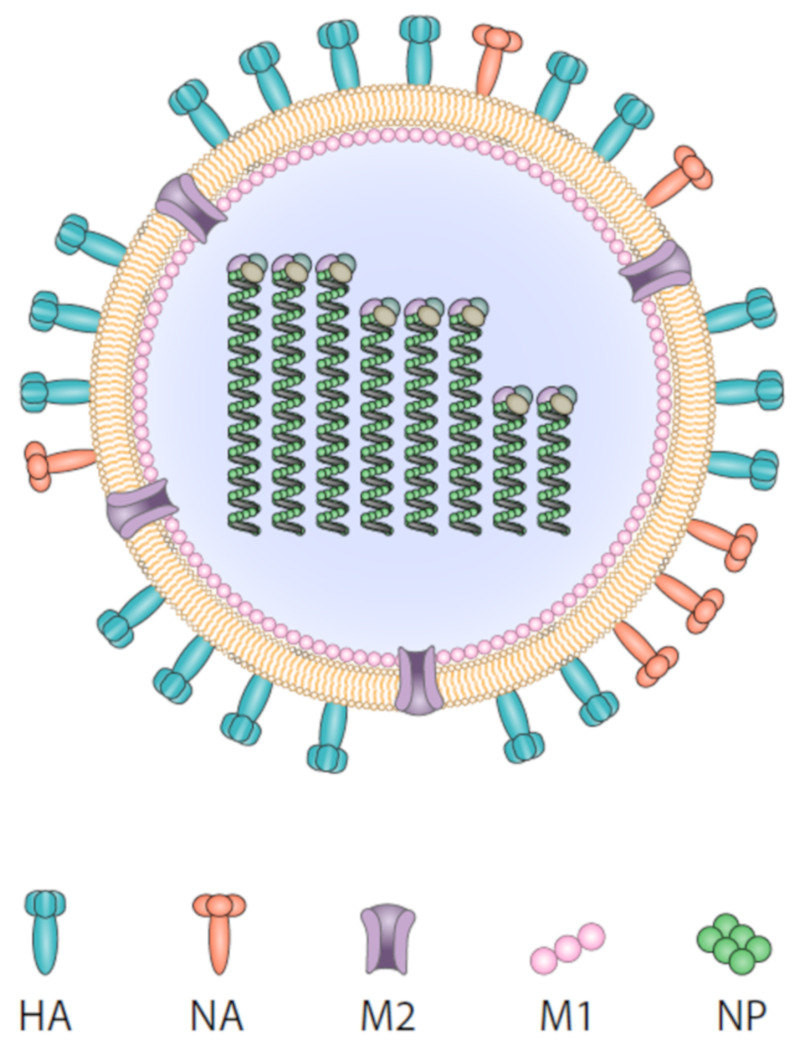
An illustration of the influenza virus and the following vaccine targets: the membrane proteins hemagglutinin (HA), neuraminidase (NA), and matrix protein 2 (M2) and the internal proteins matrix protein 1 (M1) and nucleoprotein (NP).

**Figure 2 vaccines-08-00361-f002:**
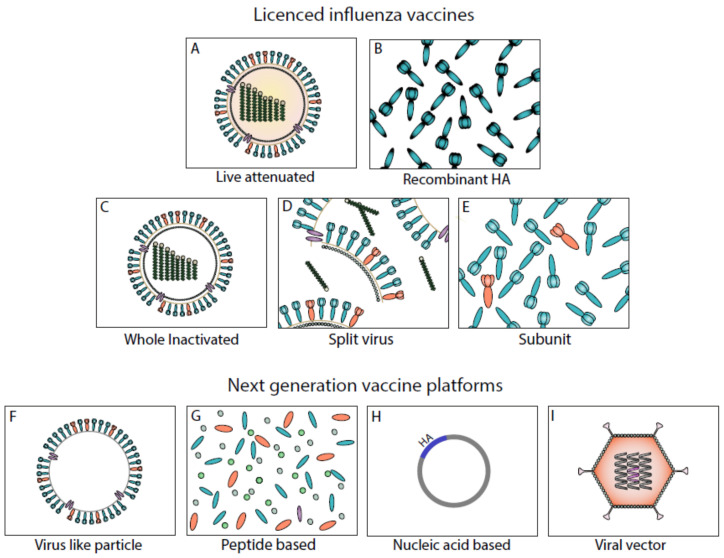
Influenza vaccine platforms: a diagram of different influenza vaccine platforms. Currently licensed vaccine formulations include live-attenuated influenza vaccines (**A**); recombinant HA vaccines (**B**); and inactivated-whole (**C**), split (**D**), or subunit (**E**) vaccines. Next-generation vaccine platforms include virus-like particles (**F**), peptide-based vaccines (**G**), nucleic-acid-based vaccines (**H**), and viral vector vaccines (**I**).

**Figure 3 vaccines-08-00361-f003:**
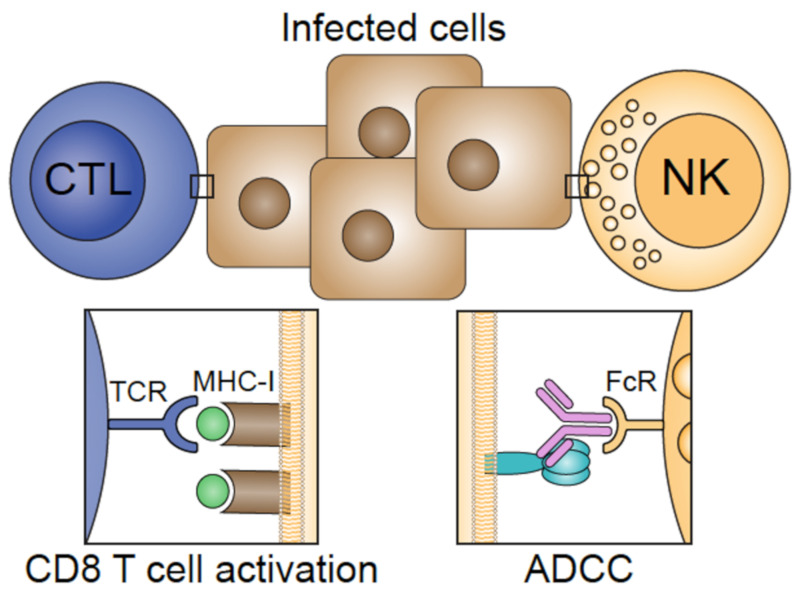
Infection-permissive immunity: T-cell receptors (TCRs) on cytotoxic T lymphocytes (CTLs) can recognize internal influenza virus proteins (NP and M1) presented by major histocompability complex (MHC ) class 1 molecules and lyse the infected host cell. Antibodies bound to viral membrane proteins presented on the surface of the host cell can contribute to clearance of infected cells by activating natural killer (NK) cells through Fc receptor (FcR) activation in a process known as antibody-dependent, cell-mediated cytotoxicity (ADCC).

**Table 1 vaccines-08-00361-t001:** Next-generation influenza vaccine attributes: the table is adapted from a report by the Center for Infectious Disease Research and Policy (CIDRAP) titled “The Compelling Need for Game-Changing Influenza Vaccines: An Analysis of the Influenza Vaccine Enterprise and Recommendations for the Future” [6].

Critical	Important	Desired
Provide protection against all HA subtypes and at a minimum protection against H1, H2, H3, H5, H7 and H9 subtypes.	Provide a decade or more of protection.	Use manufacturing technology that can be readily transferred to developing-world countries.
Provide immunologic protection for those populations most at risk for severe disease and increased mortality	Use inexpensive manufacturing technology that permits rapid and highly scalable production, particularly to address emergence of a pandemic virus.	Offer heat stability, thereby eliminating the need to maintain a cold chain.
Rarely cause adverse events, and any adverse events are mild and temporary.	Do not require injection for administration.	Do not require injection for administration.

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
