# Peer review of "Prospects and Challenges in the Development of Universal Influenza Vaccines"

_vaccines, 2020, doi:10.3390/vaccines8030361_

Round 1

Reviewer 1 Report

The authors have done a good job summarizing the current status of and the challenges facing the development of a universal influenza vaccine.

Author Response

We thank the reviewer for their positive comments.

Reviewer 2 Report

The authors discussed a topic which has been extensively discussed before, while it may represents an updates, the article require further updates on the current status of universal influenza vaccines, more details about the CMI influenza vaccines.

English Editing is required by a native English speaker as well to correct several flaws in the text. 

Best 

Author Response

Reviewer #2:The authors discussed a topic which has been extensively discussed before, while it may represents an updates, the article require further updates on the current status of universal influenza vaccines, more details about the CMI influenza vaccines.

Response: We agree with the reviewer and have now added more information about CMI influenza vaccines under “universal vaccine strategies” (lines 211-221).

English Editing is required by a native English speaker as well to correct several flaws in the text.

A native English speaker has read the manuscript and the changes are shown as track changes in the manuscript.

Reviewer 3 Report

This is a straightforward and well-written review article. As the authors highlighted, a good progress has been made in universal flu vaccine development which can and has informed the scientific community to develop vaccines for other pathogens, including SARS-CoV-2. The authors have rightly raised the issues like lack of appropriate correlation of protection and immune history that is limiting the fast-track development of universal flu vaccines. I have few minor suggestions:

  1. As the host-factors including age and sex affect influenza vaccine efficacy, universal influenza vaccine design and testing should consider these facts. I would like to suggest the authors to add this fact in the manuscript.
  2. It will be better if the authors modify the figure 3 to make the mechanisms like ADCC function and T-cell activity clearer.
  3. Minor English checks: line 8 - remove are; line 33 - remove to; line 63 - define IIVs; line 78 - better to split this paragraph to keep T-cell-based vaccines separately; line 140 - remove such as; and line 208 - remove in.  

Author Response

Reviewer #3: This is a straightforward and well-written review article. As the authors highlighted, a good progress has been made in universal flu vaccine development which can and has informed the scientific community to develop vaccines for other pathogens, including SARS-CoV-2. The authors have rightly raised the issues like lack of appropriate correlation of protection and immune history that is limiting the fast-track development of universal flu vaccines. I have few minor suggestions:

  • As the host-factors including age and sex affect influenza vaccine efficacy, universal influenza vaccine design and testing should consider these facts. I would like to suggest the authors to add this fact in the manuscript.

Response: We thank the reviewer for this helpful suggestion. We have now commented this in the manuscript (lines 231-232).

  • It will be better if the authors modify the figure 3 to make the mechanisms like ADCC function and T-cell activity clearer.

Response: We thank the reviewer for this suggestion. We have simplified figure 3 to highlight ADCC function and T-cell functions and added a few details to make the mechanisms clearer. The figure legend has been modified accordingly.

Minor English checks: line 8 - remove are; line 33 - remove to; line 63 - define IIVs; line 78 - better to split this paragraph to keep T-cell-based vaccines separately; line 140 - remove such as; and line 208 - remove in.  

Response: We apologize and thank the reviewer for pointing this out. The errors have been corrected.